# Comparative Proteomic Analysis of Exosomes Derived from Patients Infected with Non-Tuberculous Mycobacterium and *Mycobacterium tuberculosis*

**DOI:** 10.3390/microorganisms11092334

**Published:** 2023-09-17

**Authors:** Li Wang, Xubin Zheng, Jun Ma, Jin Gu, Wei Sha

**Affiliations:** 1Clinic and Research Center of Tuberculosis, Shanghai Pulmonary Hospital, School of Medicine, Tongji University, Shanghai 200433, China; 2Department of Tuberculosis, Shanghai Pulmonary Hospital, School of Medicine, Tongji University, Shanghai 200092, China; 3Shanghai Key Laboratory of Tuberculosis, Shanghai Pulmonary Hospital, School of Medicine, Tongji University, Shanghai 200092, China

**Keywords:** proteome, exosome, non-tuberculous mycobacterium, *Mycobacterium tuberculosis* complement

## Abstract

The non-tuberculous mycobacterium (NTM) is a very troublesome opportunistic pathogen, placing a heavy burden on public health. The pathogenesis of NTM pulmonary infection is not well-revealed yet, and its diagnosis is always challenging. This study aimed to use a comprehensive proteomics analysis of plasma exosomes to distinguish patients with rapidly growing NTM *M. abscessus* (MAB), slowly growing NTM *M. intracellulare* (MAC), and *Mycobacterium tuberculosis* (MTB). The identified protein components were quantified with label-free proteomics and determined with a bioinformatics analysis. The complement and coagulation were significantly enriched in patients with *Mycobacterium* infection, and a total of 24 proteins were observed with up-regulation, which included C1R, C1S, C2, MASP2, C4B, C8B, C9, etc. Of them, 18 proteins were significantly up-regulated in patients with MAB, while 6 and 10 were up-regulated in patients with MAC or MTB, respectively. Moreover, MAB infection was also related to the HIF-1 signaling pathway and phagosome processes, and MTB infection was associated with the p53 signaling pathway. This study provided a comprehensive description of the exosome proteome in the plasma of patients infected with MAB, MAC, and MTB and revealed potential diagnostic and differential diagnostic markers.

## 1. Introduction

A non-tuberculous mycobacterium is emerging worldwide as an important cause of chronic pulmonary infection [1]. Population-based studies between 2006 and 2012 from North America have documented a prevalence of 5 to 10 per 100,000 people [2]. NTM represent more than 190 species and subspecies, some of which can produce disease in humans and can affect both pulmonary and extrapulmonary sites [3]. The diagnosis of NTM pulmonary disease is very complex and relies on clinical, radiographic, and microbiologic criteria, and two or more sputum cultures are required [3]. Another challenge is that NTM pulmonary infections are often misdiagnosed as tuberculosis caused by MTB [4]. Therefore, it is particularly urgent to improve the understanding of the pathogenesis of NTM and find diagnostic markers for the diagnosis of NTM and the differential diagnosis of NTM from MTB.

An exosome (size: 30–150 nm) is a double-layered vesicular body secreted by different living cells, which exists in many physiological fluids and plants [5,6]. Studies have shown that specific cellular components in exosomes play a role in regulating intercellular communication. Proteins, metabolites, and nucleic acids delivered by exosomes effectively alter the biological response of the recipient cell. Through this mechanism, exosome-mediated responses can be either disease-promoting or disease-inhibiting. Exosomes are present in all biological fluids, making them attractive as diagnostic markers of disease that can help track disease progression. In addition, the assay of exosomes provides simultaneous access to complex extracellular and intracellular molecules, including proteins, lipids, and nucleic acids, enabling comprehensive, multiparameter diagnostic testing [7]. Exosomes were reported to be a promising research target for the diagnosis and therapy in the infection of tuberculosis [8]. Exosomes from MTB-infected host cells (e.g., macrophages and natural killer cells) trigger different immune responses such as inflammatory responses, antigen presentation, and activation of subsequent pathways, highlighting the key function of exosomes in the immune response against MTB [9]. However, there are few reports about the function of exosomes derived from patients with NTM pulmonary infection, leaving the role of exosomes in the development of NTM pulmonary infection unclear.

NTM are classified into two categories based on the interval to colony formation by the subculture on solid media: rapidly growing NTM (RGM) and slowly growing NTM (SGM) [10]. RGM and SGM are two completely different types of NTM in terms of causing disease progression in infected patients. In general, pulmonary infections caused by SGM, e.g., *M. avium*, MAC, and *M. kansasii*, progress relatively slowly, while those caused by RGM, e.g., MAB, progress significantly more rapidly. MAB and MAC are the most common RGMs and SGMs in clinical practice, respectively, and therefore, they were used as the subjects of the study. In this study, we report for the first time a preliminary screening proteomics analysis of plasma exosomes derived from healthy controls, patients with MAB, patients with MAC, and patients with MTB. Our study aimed to provide clues for the pathogenesis of NTM and MTB infections, and to provide potential diagnostic markers for their diagnosis and differential diagnosis. The preliminary results merit validation with a large sample to guide clinical practice.

## 2. Methods

### 2.1. Grouping and Ethics Statement

In the present study, a total of 30 patients admitted to Shanghai Pulmonary Hospital (SPH) between 2021 and 2022 were recruited (Figure 1A), of which 10 patients were diagnosed with pulmonary MAB infection (MAB group), 10 patients with pulmonary MAC infection (MAC group), and 10 patients with active MTB infection (MTB group). The ten patients in the MTB group were all diagnosed with drug-sensitive active pulmonary TB infection. The following are the inclusion criteria: (1) the diagnosis is clear (two positive sputum cultures with the bacterial species well-defined); (2) pulmonary infection; (3) age between 18 years old to 75 years old; (4) immunocompetent—evaluation based on routine blood count and basic flow cytometry to obtain lymphocyte subset counts; and (5) the patients’ first diagnosis was NTM or MTB, and all of them did not receive treatment upon enrollment. The exclusion criteria are as follows: (1) patients with NTM and MTB combined infection; (2) patients with more than one NTM species infection; (3) patients with extrapulmonary NTM or MTB infection; (4) patients with diabetes, hyperlipidemia, and other metabolic diseases; (5) HIV positive; (6) patients with another infection requiring treatment; and (7) patients needing oxygen therapy because of severe lung infection. The healthy control (HC) group included 10 volunteers who took routine physical examinations in SPH. The clinical characteristics of patients and healthy controls are provided in Appendix A. All patients were diagnosed according to a combination of clinical, radiographic, and microbiologic criteria [3,11]. Before treatment, blood (5 mL per donor) was collected into heparin-containing tubes and centrifuged at 2000× *g* for 10 min to obtain plasma.

The present study was approved by the Ethics Committee of Shanghai Pulmonary Hospital (Shanghai, China; approval no. K21-257). The study was performed in accordance with the Declaration of Helsinki Principles and all donors provided written informed consent to participate in the study.

### 2.2. Isolation of Exosomes

Exosomes were isolated via ultracentrifugation. In the first step, centrifugation was performed at 2000× *g* for 30 min and 10,000× *g* for 45 min to remove cellular debris and larger vesicles. Subsequently, the supernatant was filtered through a 0.45 μm membrane and the filtrate was collected. Finally, after 100,000× *g* centrifugation for 70 min, the supernatant was removed to obtain the final exosome. All operations were carried out at 4 °C. The purified exosomes were then identified with a transmission electron microscopy analysis, nanoparticle size analysis, and nanoflow analysis.

### 2.3. Transmission Electron Microscopy Analysis

After resuspension in PBS, the exosome dilution solution (10 µL) was dropped on a copper wire at 25 °C and washed three times with ultrapure water. After air-drying, 10 µL of a phosphotungstic acid solution was added on copper mesh for 1 min. Samples were analyzed using a Hitachi HT-7700 transmission electron microscope (Hitachi, Ltd., Tokyo, Japan) to observe exosome morphology.

### 2.4. Nanoparticle Size Analysis

The exosomes were resuspended with 100 μL of a pre-cooled PBS solution, and 10 µL of the resuspension was diluted with ultrapure water to 1 mL. After the performance of the instrument was tested as qualified using the standard substance, the exosome sample was analyzed with a NanoFCM N30E Flow NanoAnalyzer (NanoFCM Co., Ltd., Nottingham, UK).

### 2.5. Nanoflow Analysis

In this study, a nanoflow analysis was used to detect exosomal markers, and nanoflow can only detect membrane surface proteins at present. CD9 and CD81 were chosen to identify exosomes, both of which are transmembrane proteins directly involved in sorting the contents of extracellular vesicles. The diluted exosomes were added with fluorescently labeled antibodies (CD9 and CD81, BD Tech, Becton Dr., Franklin Lakes, NJ, USA), mixed evenly, and incubated at 37 °C for 30 min in the dark. Then, the exosomes were subsequently washed twice with a pre-cooled 1 mL PBS solution (4 °C, 110,000× *g*, 70 min). The supernatant was removed, and the exosomes were resuspended with a 50 μL pre-cooled PBS solution. After the performance of the instrument was tested as qualified using the standard substance, the exosome sample was analyzed with a NanoFCM N30E Flow NanoAnalyzer (NanoFCM Co., Ltd., Nottingham, UK).

### 2.6. Extraction of Exosomal Protein

Exosomal protein was extracted with lysates containing 8 M of urea/50 mM of Tris-HCL and a Roche cocktail [12]. Exosomal protein was quantified with the Bradford method and subjected to SDS-PAGE electrophoresis for a quality control analysis. Then, the Trypsin enzyme was added in the ratio of protein/enzyme = 50:1 for enzymolysis for 16 h at 37 °C. The enzymolyzed peptide segments were desalted by using a Waters solid phase extraction column and were pumped dry in a vacuum. The extracted peptide segments were reconstituted with pure water and stored at −20 °C.

### 2.7. Nano-LC-MS/MS

The peptides were separated on a Thermo Scientific UltiMate™ 3000 Binary Rapid Separation System (Thermo Fisher Scientific, San Jose, CA, USA) with a 3.5 μm 4.6 × 150 mm Agilent ZORBAX 300Extend-C18 column for liquid phase separation of the samples. The eluted peaks were monitored at 214 nm and one component was collected per minute. The samples were combined with the chromatographic eluted peaks to obtain 10 components, which were then lyophilized. The lyophilized peptides were re-dissolved, centrifuged at 20,000× *g* for 10 min, and the supernatant was sampled. Subsequently, the samples were separated with the Thermo Scientific EASY-nLC™ 1200 system (Thermo Fisher Scientific, San Jose, CA, USA). The peptide fragments separated in the liquid phase were ionized with the nanoESI source and then entered the mass spectrometer Orbitrap Exploris 480 (Thermo Fisher Scientific, San Jose, CA, USA) for DDA (data-dependent acquisition) mode detection. Two QC samples were inserted before, during, and after the whole experiment to evaluate the stability and repeatability of the experiment.

### 2.8. Protein Identification and Quantification

The protein search identification and quantitation of raw data from the mass spectrum of label-free DDA were performed using MaxQuant software (https://www.maxquant.org/ (accessed on 3 December 2022), Version 2.1.4.0). The main parameters of MaxQuant were as follows: the mass spectrometric type was set to standard mode, Trypsin was the Trypsin/P enzyme, and the enzyme was fixed and modified to Carbomidomethyl (C), and variably modified to Oxidation (M) and Acetyl (protein N-term). A match between runs and secondary peptide search were also checked, and other parameters were the default parameters. Identification conditions are as follows: (1) chromatogram false positive (PSM FDR) < 0.01, and (2) removal of proteins from reverse and contaminant libraries. The database used for this protein identification was the human Uniprot database (20,399 sequences in total, https://www.uniprot.org/ (accessed on 3 December 2022)). Quality control information for peptide identification is provided in Appendix A.

### 2.9. Statistical Analysis

The statistical analysis was mainly performed with R software (version 4.0). The original intensity values of the proteins were normalized with the medium, and the clustering heat map was drawn with the R-package phenomap (https://cran.r-project.org/web/packages/phenomap/ (accessed on 10 December 2022)). A PCA analysis and significantly different protein analysis were performed with the R-package metaX [13]. A T test was used for a statistical differential analysis and a cut-off *p*-value ≤ 0.05 and fold change ≥ 1.5 were used to select statistically differentially expressed proteins. The Gene Ontology (GO, http://geneontology.org/ (accessed on 10 December 2022)) term, Kyoto Encyclopedia of Genes and Genomes (KEGG, https://www.kegg.jp/ (accessed on 10 December 2022)), pathway was used to enrich the functional items of differential proteins, and a cut-off *p*-value < 0.05 was used. Subcellular localization of the protein was performed with software WoLF PSORT (https://wolfpsort.hgc.jp/ (accessed on 10 December 2022)).

## 3. Results

### 3.1. Characterization of Plasma Exosomes

According to the TEM analysis, the exosomes were typical cup-shaped vesicles, and no shape difference was observed among the exosomes from the plasma of healthy controls, patients with NTM, and patients with MTB (Figure 1C). The nanoparticle size analysis showed a particle size distribution between 40 and 150 nm, and the particle size of healthy controls, patients with NTM, and active-MTB-infected patients was 83.75 nm, 85.22 nm, and 84.7 nm, suggesting that these particles were exosomes (Figure 1D). The nanoflow analysis showed that the exosomes were positive for CD9 and CD81 (Figure 1E). Overall, high-quality exosomes were obtained with ultracentrifugation, which provided the basis for subsequent exosome proteomics.

### 3.2. Up-Regulation of Complement Proteins in Plasma Exosomes from NTM-Infected Patients

Since RGM infection and SGM infection progress differently, it is important to separate MAB-infected patients from MAC-infected patients and compare them with healthy controls separately. Firstly, plasma exosome proteins from the MAB group and MAC group were compared with those from the HC group, respectively, to reveal the pathogenesis of these two infectious diseases and potential diagnostic markers.

Between the MAB group and HC group, there were 139 differentially expressed proteins, including 127 up-regulated and 12 down-regulated proteins, represented with a histogram, volcano plot, intensity mean distribution, heatmap, and PCA analysis (Figure 1B and Figure 2A–C). According to the results of the GO term enrichment analysis, the differential expression proteins were mainly involved in biological processes such as regulation of the humoral immune response, regulation of complement activation, and the humoral immune response mediated with circulating immunoglobulin (Figure 2D).

Furthermore, the results of the KEGG pathway enrichment analysis showed that complement and coagulation cascades was significantly enriched (Figure 2E), with 18 proteins significantly up-regulated, including C1R, C1S, C2, MASP2, C4B, CFI, C9, CLU, CFHR3, SERPING1, KNG1, F5, FGA, FGG, SERPINA1, SEPRINA5, SEPRND1, and SEPRINF2 (Figure 3A). ROC revealed that all these 17 complement and coagulation cascade proteins except MASP2 are potential diagnostic exosomal markers for MAB with AUC > 0.80 (Figure 3B). In addition, the HIF-1 signaling pathway was also enriched, with PGK1, TIMP1, and AKT3 significantly up-regulated in the MAB group (Figure 2F). In addition, phagosome processes and tuberculosis disease were also significantly enriched, and AKT3, CD14, FCGR3A, and CORO1A were significantly up-regulated in the MAB group (Figure 2G). Notably, nitrogen metabolism was also enriched, with CA2 and CA4 significantly up-regulated (Figure 2H). The ROC analysis revealed that AKT3, CD14, and FCGR3A are potential diagnostic exosomal protein markers for MAB-infected patients with AUC > 0.80 (Figure 2I).

Between the MAC group and HC group, there are 97 differential expression proteins, including 88 up-regulated proteins and 9 down-regulated proteins (Figure 1B). Based on the results of the GO analysis, the differential expression proteins were mainly involved in biological processes such as wound healing, positive regulation of the response to a stimulus, and the inflammatory response (Figure 4D). Results of the KEGG analysis showed that complement and coagulation cascades were also significantly enriched (Figure 4E). Six proteins in pathways of complement and coagulation cascades were significantly up-regulated, including CLU, F5, FGA, SERPINA1, SERPINA5, and SERPING1 (Figure 4F). The ROC analysis revealed that CLU, F5, FGA, SERPINA1, and SERPINA5 are potential diagnostic protein markers for MAC-infected patients with AUC > 0.80 (Figure 4G).

### 3.3. More Up-Regulated Complement Proteins in MAB-Infected Patients than MAC

Subsequently, plasma exosomal proteins from the MAB group and MAC group were compared to reveal the differences and similarities of the pathogenesis of these two infectious diseases and to screen for potential differential diagnostic markers.

The results showed that there are 25 differential expression proteins between the MAB group and MAC group, including 21 up-regulated proteins and 4 down-regulated proteins. Based on the results of the GO analysis, the differential expression proteins between the MAB group and MAC group were mainly involved in biological processes such as complement activation, the humoral immune response mediated with circulating immunoglobulin, regulation of the humoral immune response, and lymphocyte-mediated immunity (Figure 5D). Interestingly, the results of KEGG pathway enrichment showed that complement and coagulation cascades were also significantly enriched between the MAB group and MAC group (Figure 5E). Complement and coagulation cascade proteins C1R, C4B, C8B, C9, CFHR3, and SERPIND1 were significantly increased in exomes from the MAB group (Figure 5F). In addition, the ROC analysis revealed that C4B, C9, and CFHR3 are able to distinguish between MAB-infected patients and MAC-infected patients with AUC > 0.80 (Figure 5G).

### 3.4. Comparison of Plasma Exosomes between NTM-Infected Patients and MTB-Infected Patients

The diagnosis and differential diagnosis between NTM infection and MTB infection are complex due to similarities in clinical presentations. It is reported that due to the inaccuracy of species identification prior to initiation of treatment, a proportion of previously diagnosed recurrent tuberculosis cases are attributed to the transition between MTB and NTM [14]. Therefore, the exosomal proteins from NTM-infected patients and MTB-infected patients were compared.

Since the clinical characteristics of SGM pulmonary infection resemble MTB [15], it was necessary to explore more sensitive and specific clinical biomarkers screened from plasma omics. Between the MAC group and MTB group, a total of 31 differentially expressed exosomal proteins were identified, including 21 up-regulated and 10 down-regulated proteins (Figure 6A–C). According to the results of the GO analysis, the differential expression proteins were mainly involved in biological processes such as regulation of the cellular component size, and maintenance of the location in the cell (Figure 6D). According to the results of the KEGG analysis, complement and coagulation cascades were also significantly enriched, with SERPINC1 and VTN decreased in MAC-infected patients (Figure 6E,F). Additionally, the tight junction and regulation of the actin cytoskeleton were also enriched, with SLC9A3R1, MSN, and TMSB4X increased in MAC-infected patients, while MSN and TMSB4X showed potential to differentiate MAC-infected patients and MTB-infected patients (Figure 6G,H).

Between the MAB group and MTB group, there are 63 differential expression proteins, including 43 up-regulated proteins and 20 down-regulated proteins (Appendix A). The results of the GO analysis showed that the differential expression proteins were mainly involved in biological processes such as regulation of very-low-density lipoprotein particle clearance, vesicle-mediated transport, regulation of the humoral immune response, and regulation of hydrolase activity. Complement and coagulation cascades were also significantly enriched between the MAB group and MTB group. Complement and coagulation cascade proteins C4B, SERPINA5, and SERPING1 were increased in the MAB group, while F11 and F13B were decreased. Additionally, gluconeogenesis was also enriched, with PGK1 and PKM increased in the MAB group. The HIF-1 signaling pathway was also enriched, with PGK1 increased and TF decreased in the MAB group. The ROC analysis revealed that SERPINA5, SERPING1, F11, F13B, PGK1, PKM, and TF are potential diagnostic exosomal protein markers to differentiate MAB-infected patients and MTB-infected patients.

Between the MTB group and HC group, there are 49 differential expression proteins, including 43 up-regulated proteins and 6 down-regulated proteins (Appendix A). According to the results of the GO analysis, the differential expression proteins were mainly involved in biological processes such as high-density lipoprotein particle remodeling, hemostasis, and wound healing. According to the results of the KEGG analysis, complement and coagulation cascades were also significantly enriched between the MTB group and HC group. SERPINC1, SERPINA1, SERPINA5, FGA, FGG, C9, CLU, MBL2, F5, and VTN were increased in the MTB group. Moreover, the p53 signaling pathway was enriched, with IGF1 and IGFBP3 increased in the MTB group. The ROC analysis revealed that IGFBP3, F5, SERPINC1, and SERPINA1 are potential diagnostic markers of MTB-infected patients.

### 3.5. Comprehensive Analysis of Pathways

Complement and coagulation cascades were enriched in all KEGG pathway analyses for differential proteins as described above, and a total of 24 differential proteins of complement and coagulation cascades were identified (Figure 7A). The majority of these proteins (18/24) were significantly up-regulated in the MAB group compared to the HC group. A total of 6 proteins in the MAC group and 10 proteins in the MTB group were up-regulated. Notably, exosomes from the MAB group had the highest levels of complement and coagulation cascade protein expression, and exosomes from the MAC group and MTB group had similar levels of complement and coagulation cascade protein expression. According to these findings, the complement and coagulation cascade was extensively and significantly up-regulated in exosomes from the MAB group, while exosomes from the MAC group and MTB group showed some resemblance in the complement and coagulation cascade activation.

The results also revealed that the HIF-1 signaling pathway, phagosome processes, tuberculosis disease, and gluconeogenesis were enriched in the KEGG pathway analysis. PGK1 (phosphoglycerate kinase 1) and PKM (pyruvate kinase M) were significantly up-regulated in the MAB group. Both proteins are involved in glycolysis, indicating that in the MAB group, cells secreting exosomes might conduct more glycolysis (Figure 7B). TIMP1 (metallopeptidase inhibitor 1), AKT3 (serine/threonine kinase 3), CD14, FCGR3A (Fc gamma receptor IIIa, CD16), and CORO1A (coronin 1A) were up-regulated in the MAB group. CD14 was also up-regulated in the MTB group and CORO1A was up-regulated in the MAC group.

## 4. Discussion

Although accumulating studies have reported to use mass spectrometry to analyze exosomal protein related to a *Mycobacterium* [8,16], there have been no studies carried out on a comprehensive analysis of plasma exosomal protein from patients with NTM pulmonary infection and patients with MTB pulmonary infection to elucidate the exosomal protein profiles of these two disease states. This is the first comprehensive protein omics analysis of exosomes from these physiological or pathophysiological states. In the present study, the exosomes were round, with a diameter of 80 to 90 nm, and expressed the characteristic proteins CD9 and CD81. Based on these results, high-purity exosomes were isolated in our experiment, which were crucial for a downstream protein omics analysis.

In the present study, a total of 24 differential proteins of complement and coagulation cascades were identified. An analysis of a proteomic dataset of exosomes revealed that proteins of a complement and coagulation cascade were significantly up-regulated in the MAB group, MAC group, and MTB group, comparing to the HC group. It is demonstrated that the activation of a complement and coagulation cascade was related to these three disease states. Moreover, the results suggested that the MAB group had greater amounts and higher levels of complement and coagulation cascade proteins than other groups. This phenomenon is most likely explained with the hypothesis that the progression of *Mycobacterial* pulmonary infection disease is associated with complement activation. To test this hypothesis, more work needs to be done in the pipeline. The results of the ROC analysis revealed the proteins of a complement and coagulation cascade are potential diagnostic biomarkers of the above three disease states, especially patients with MAB infection.

The complement system is an important part of the innate and adaptive immune systems, which enhances the function of antibodies and phagocytes [17,18]. The complement system plays a decisive role in the defense against pathogenic microbial infection [19]. Macrophages could engulf mycobacterial bacilli through bacterial cell surface proteins or secreted proteins and activate the complement pathway [20]. The classical pathway is activated by C1q, which binds to the antibody–antigen complex and subsequently activates complement pathway proteins such as C1R and C1S, as well as C4B. Mannan-binding lectin-associated serine protease (MASP), a multifunctional serine protease, plays a key role in the activation of the complement lectin pathway [21]. Together with alternative pathways, all three complement pathways contribute to an immune response involved in eliminating mycobacteria [20]. The complement system plays a very critical role in the host’s defense of mycobacteria. It is reported that deficient complement opsonization could impair *Mycobacterium avium* killing with neutrophils in cystic fibrosis [22]. In a multiple-cohort study of active tuberculosis, increased expression of SERPING1 was identified in patients with active tuberculosis [23]. In addition, C1q was positively correlated with C1-inhibitor levels and the combined increase in C1q and C1-inhibitor levels was highly specific for active tuberculosis. These results are in good agreement with our study. A study by Lubbers et al. [24] also identified the complement component C1q as a serum biomarker for the detection of active tuberculosis. Moreover, in a study of serum exosomes in patients with active MTB infection, the complement component C9 was significantly up-regulated [8].

Furthermore, our analysis revealed that the HIF-1 signaling pathway was also enriched in MAB-infected patients compared to healthy controls. Hypoxia-inducible factor 1 (HIF-1) is an oxygen-regulated transcription activator, which plays an important role in mammalian development, physiology, and disease pathogenesis [25]. Up-regulation of HIF-1 and a metabolic reprogramming to the Warburg Effect-like state are known to be critical for immune cell activation in response to MTB infection [26]. In response to infection, the activation of host innate immune and adaptive immune cells is accompanied by the transformation of the biological energy pathway from oxidative phosphorylation to glycolysis, a metabolic remodeling called the Warburg effect, which is necessary to produce antibacterial and pro-inflammatory effector molecules. The underlying mechanism of the Warburg effect focuses on the expression and regulation of hypoxia-inducible factor 1α (HIF-1α) [27]. It is demonstrated that activation of HIF-1 could enhance bactericidal effects of macrophages to MTB [28]. A study conducted by Teran et al. [29] also demonstrated in an in vitro and in vivo experiment that MTB infection stimulates the expression of HIF-1, which leads to macrophage activation.

Interestingly, phagosome processes and tuberculosis disease were also significantly enriched, with AKT3, CD14, FCGR3A, and CORO1A being significantly up-regulated in MAB-infected patients. In addition, PGK1 and PKM were also significantly up-regulated in MAB-infected patients. PGK1 is a glycolytic enzyme that catalyzes the conversion of 1,3-diphosphoglycerate to 3-phosphoglycerate. PKM is a pyruvate kinase that catalyzes the transfer of a phosphoryl group from phosphoenolpyruvate to ADP, generating ATP and pyruvate. Both enzymes are key enzymes in glycolysis. In this study, these two enzymes were significantly increased in MAB-infected cells, suggesting that glycolysis was more active in the cells of MAB-infected patients. This phenomenon might also be related to the Warburg effect mentioned above, that is, the activation of the host’s innate immune and adaptive immune cells in response to infection is accompanied by the transformation of the bioenergy pathway from oxidative phosphorylation to glycolysis.

Currently, there is still considerable work to be done on the detailed function of exosomes secreted by *Mycobacterium*-infected cells. A recent study demonstrated that exosomes derived from MTB-infected Mesenchymal stem cells induce a pro-inflammatory response of macrophages through elevating the production of TNF-α, RANTES, and iNOS [30].

Due to limited specimens, only exosomal proteins from *Mycobacterium*-infected patients were analyzed in the present study. However, both lipids and nucleic acids in exosomes are also very important. Nucleic acids in exosomes, especially non-coding RNAs, are very important in mycobacterial infections. A recent study performed immense parallel sequencing for exosomal ncRNAs and finally collected miR-185-5p as a promising potential biomarker for tuberculosis [31].

The present study also has some other shortcomings. Notably, due to the precious and rare samples, this study is only a pioneering and exploratory screening study based on LC-MS, and validation work is absent. Therefore, as a next step, there will be continued research on NTM exosomes, and more cases will be recruited to validate the exosomal proteomics. In addition, the specific molecular mechanisms of NTM or MTB infection identified in this study require a further analysis with laboratory experiments.

## 5. Conclusions

Based on preliminary proteomic screening, MAB infection, MAC infection, and MTB infection were all associated with increased complement proteins in the patient’s plasma exosomes, with MAB infection accounting for the highest increase in complement proteins. MAB infection is also related to the HIF-1 signaling pathway, phagosome processes, and glycolysis. Our research provided vital clues for the function of exosomal proteins during NTM infection and provided potential exosome diagnostic and differential diagnostic markers for NTM infection and MTB infection.

## Figures and Tables

**Figure 1 microorganisms-11-02334-f001:**
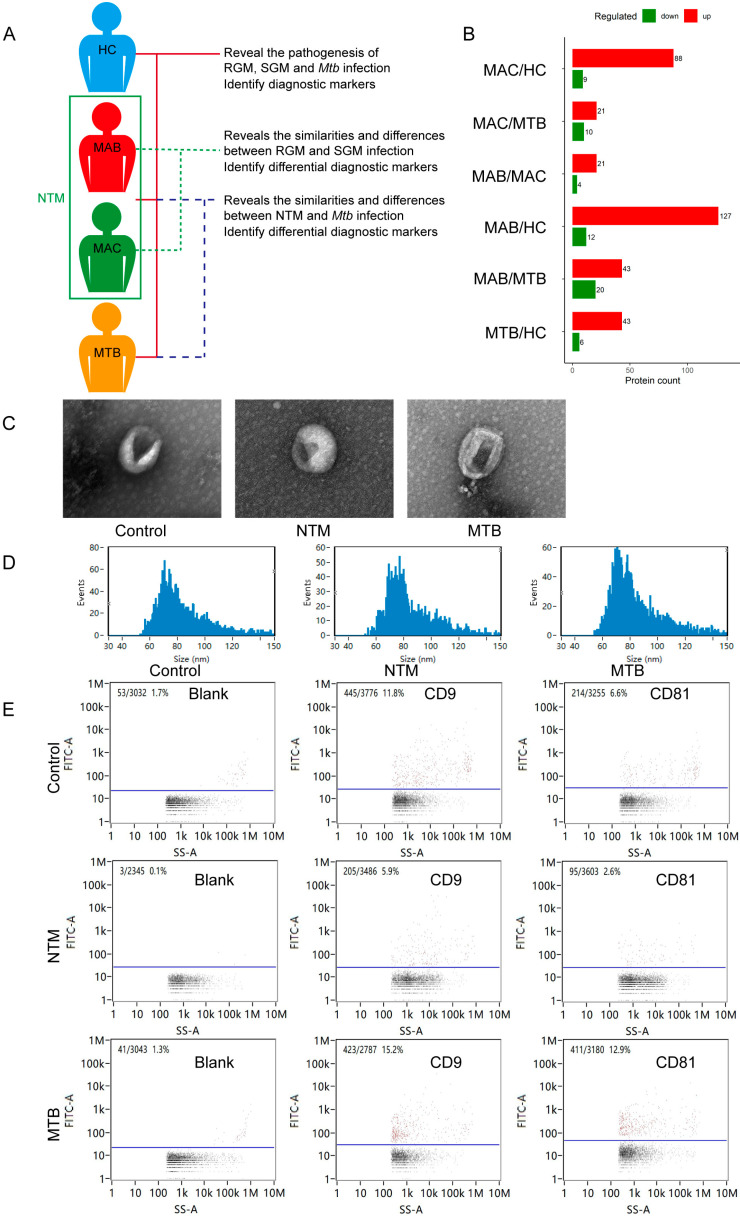
Schematic diagram and biological characterization of exosomes. (**A**) Schematic diagram of proteomic analysis of plasma exosomes from healthy controls, NTM-infected patients, and MTB-infected patients. (**B**) Histogram of the number of differential proteins between groups. (**C**) Transmission electron micrographs of isolated exosomes. Representative image of exosomes at a magnification of × 60,000. (**D**) Size distribution of exosomes. Particle size of healthy controls, NTM-infected patients, and MTB-infected patients was 83.75 nm, 85.22 nm, and 84.7 nm. Particle concentrations of healthy controls, NTM-infected patients, and MTB-infected patients were 3.34 × 10^9^, 2.57 × 10^10^, and 1.68 × 10^9^ particles/mL. (**E**) Exosomal marker proteins CD9 and CD81. The red dots represent exosomes.

**Figure 2 microorganisms-11-02334-f002:**
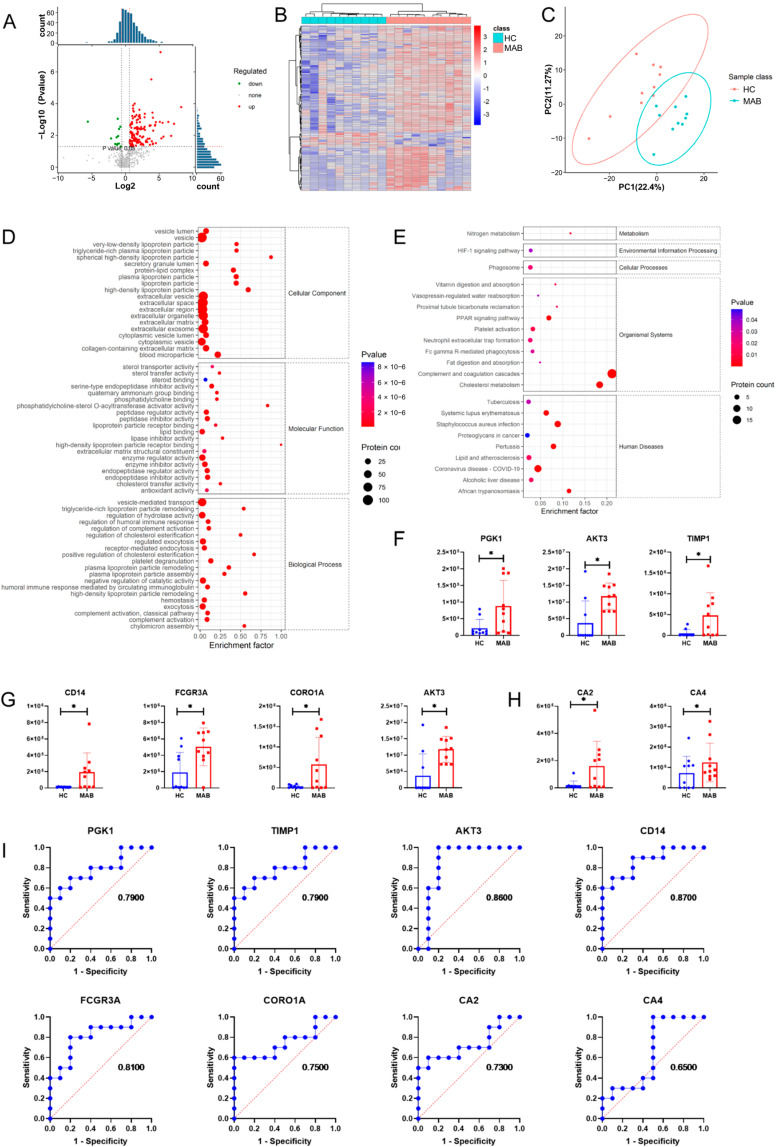
Proteomics analysis of plasma exosomes between MAB group and HC group. (**A**) Volcano plot of differential expression proteins. (**B**) Heatmap of differential expression proteins. (**C**) PCA plot of expression plasma exosomal proteins. (**D**) GO term enrichment map of differential expression proteins. (**E**) KEGG pathway enrichment map of differential expression proteins. (**F**) Histogram of differential protein expression of HIF-1 signaling pathway. (**G**) Histogram of differential protein in tuberculosis disease. (**H**) Histogram of differential protein in nitrogen metabolism. (**I**) ROC analysis of 8 differential expression proteins. * *p* < 0.05.

**Figure 3 microorganisms-11-02334-f003:**
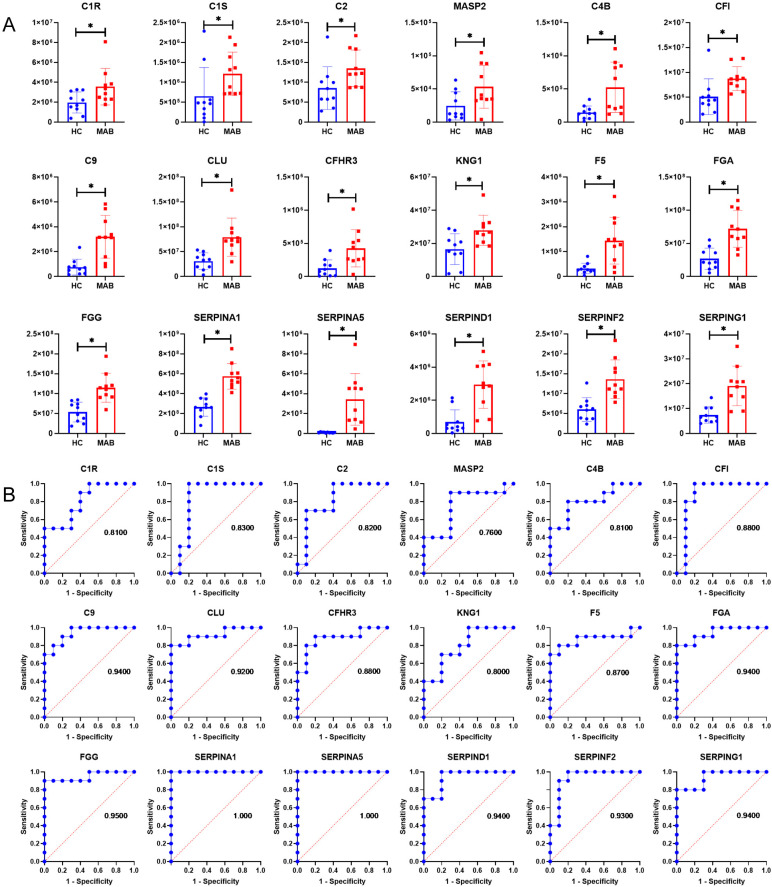
Complement and coagulation cascade differential expression proteins between MAB group and HC group. (**A**) Histogram of differential protein expression in complement and coagulation cascade. (**B**) ROC analysis of complement and coagulation cascade differential expression proteins. * *p* < 0.05.

**Figure 4 microorganisms-11-02334-f004:**
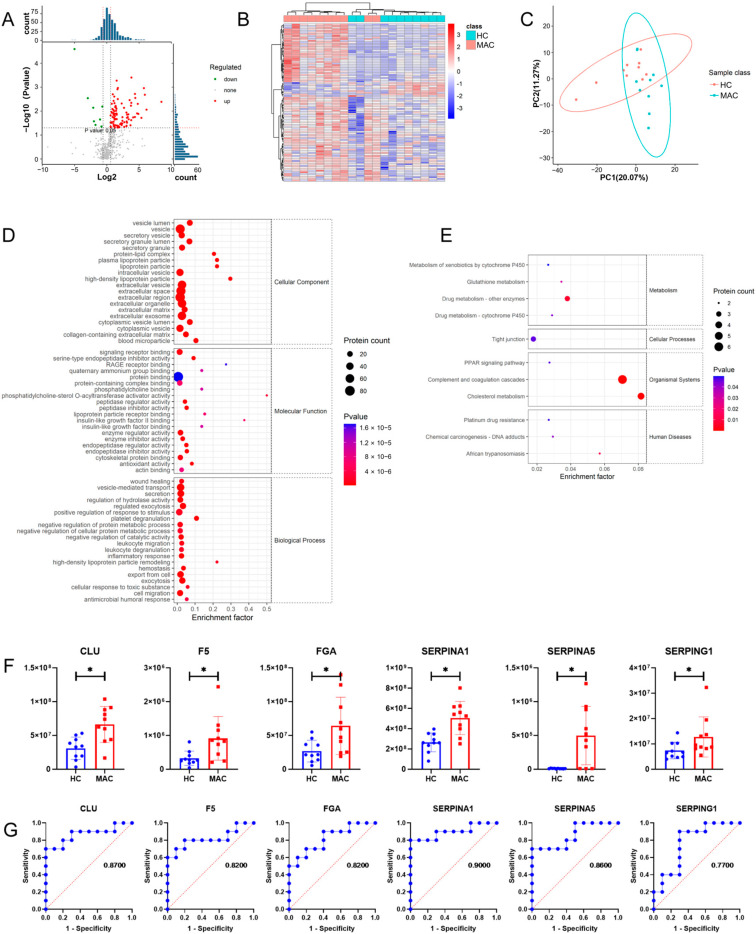
Proteomics analysis of plasma exosomes between MAC group and HC group. (**A**) Volcano plot of differential expression proteins. (**B**) Heatmap of differential expression proteins. (**C**) PCA plot of expression plasma exosomal proteins. (**D**) GO term enrichment map of differential expression proteins. (**E**) KEGG pathway enrichment map of differential expression proteins. (**F**) Histogram of differential protein expression in complement and coagulation cascades. (**G**) ROC analysis of complement and coagulation cascade differential expression proteins. * *p* < 0.05.

**Figure 5 microorganisms-11-02334-f005:**
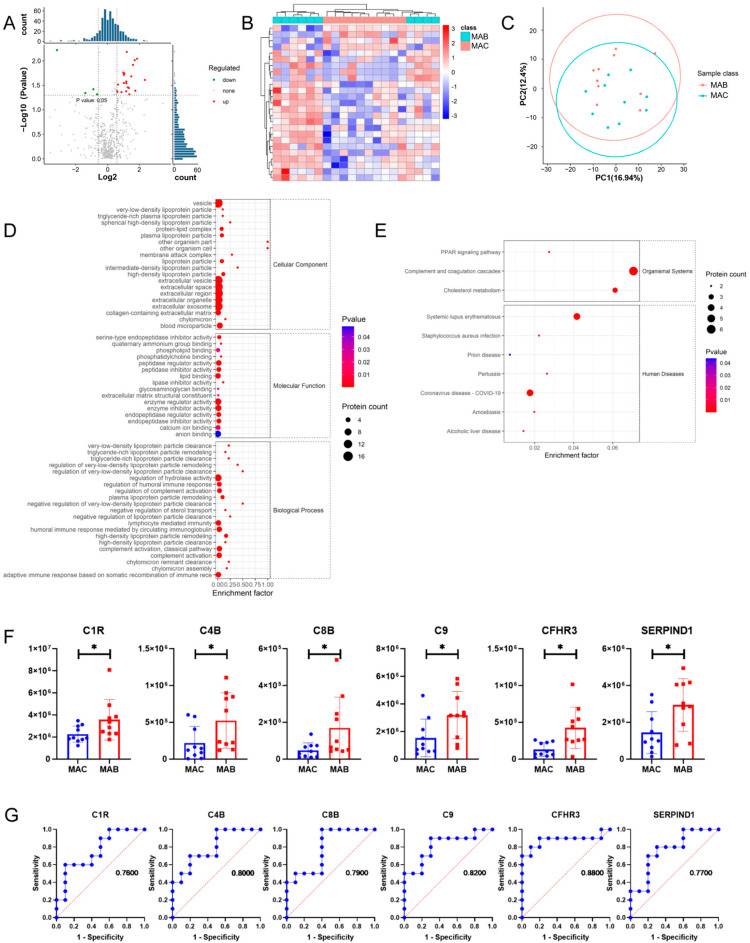
Proteomics analysis of plasma exosomes between MAB group and MAC group. (**A**) Volcano plot of differential expression proteins. (**B**) Heatmap of differential expression proteins. (**C**) PCA plot of expression plasma exosomal proteins. (**D**) GO term enrichment map of differential expression proteins. (**E**) KEGG pathway enrichment map of differential expression proteins. (**F**) Histogram of differential protein expression in complement and coagulation cascade. (**G**) ROC analysis of complement and coagulation cascade differential expression proteins. * *p* < 0.05.

**Figure 6 microorganisms-11-02334-f006:**
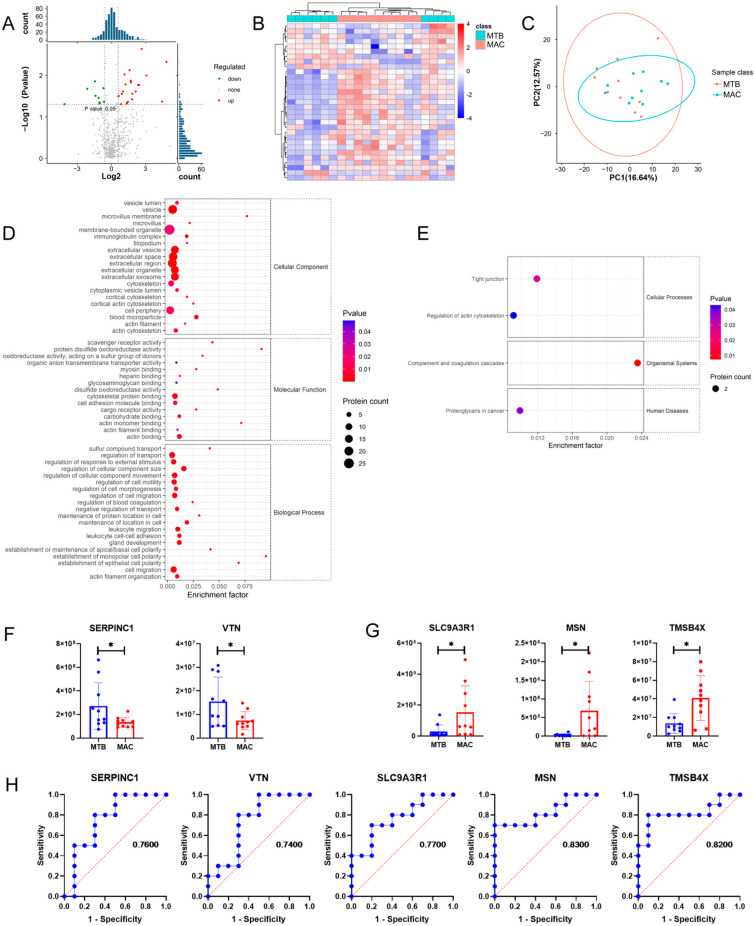
Proteomics analysis of plasma exosomes between MAC group and MTB group. (**A**) Volcano plot of differential expression proteins. (**B**) Heatmap of differential expression proteins. (**C**) PCA plot of expression plasma exosomal proteins. (**D**) GO term enrichment map of differential expression proteins. (**E**) KEGG pathway enrichment map of differential expression proteins. (**F**) Histogram of differential protein expression in complement and coagulation cascades. (**G**) Histogram of differential protein expression in tight junction and regulation of actin cytoskeleton. (**H**) ROC analysis of differential expression proteins. * *p* < 0.05.

**Figure 7 microorganisms-11-02334-f007:**
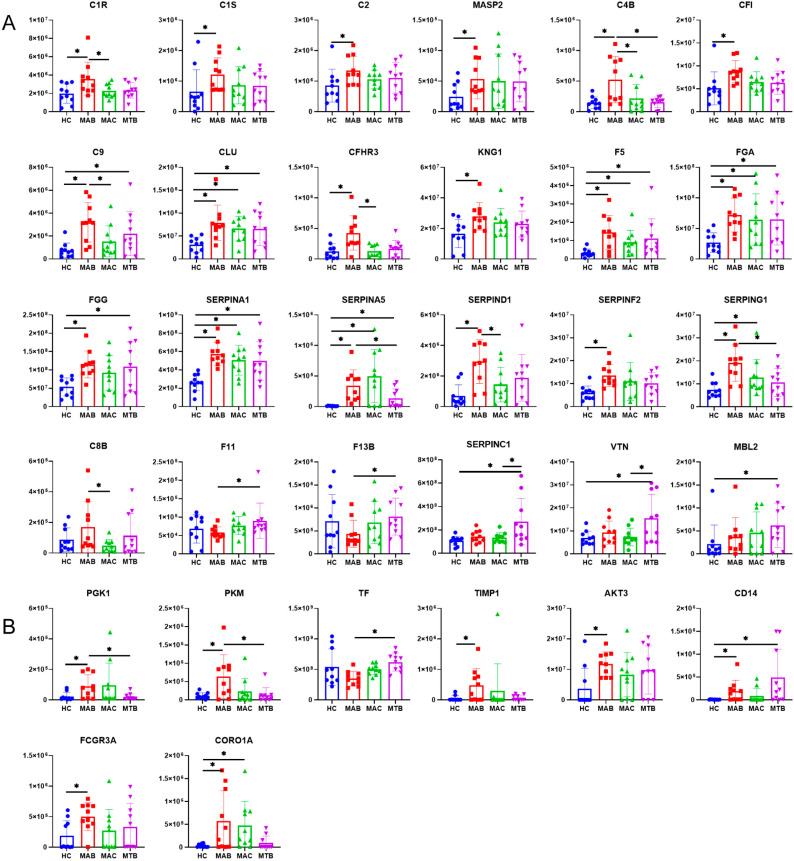
Histogram of differential protein expression among four groups. (**A**) Histogram of differential expression protein in complement and coagulation cascades. (**B**) Histogram of differential expression protein in HIF-1 signaling pathway, phagosome processes, tuberculosis disease, and gluconeogenesis. * *p* < 0.05.

## Data Availability

The datasets used and/or analyzed during the current study are available from the corresponding author on reasonable request.

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
