# Peer review of "Comparative Proteomic Analysis of Exosomes Derived from Patients Infected with Non-Tuberculous Mycobacterium and Mycobacterium tuberculosis"

_microorganisms, 2023, doi:10.3390/microorganisms11092334_

Round 1

Reviewer 1 Report

Wang et al. have conducted a commendable preliminary investigation into exosome proteomics analysis to distinguish between NTM and MTB, building upon prior MTB research. This preliminary data holds promise for publication.

However, the absence of validation for the KEGG pathway analysis raises significant concerns about the data's reliability. To solidify your findings, it is essential to perform quantitative western blot or a similar proteomics analysis. Select key proteins from both the downregulated and upregulated categories and confirm your results using an independent secondary method. This may entail substantial work in the lab, but it is crucial for ensuring the quality of your project.

Additional comments:

Throughout your manuscript, you consistently refer to NTM as SGM and RGM collectively, suggesting common findings. However, your study cohorts exclusively involve M. Abscesses and M. Intracellulare. Therefore, it's necessary to narrow your findings to only these two species, as you acknowledged in the introduction. With over 190 NTM species reported, it is not accurate to generalize common findings without supporting data.

Minor comments:

1. Line 45 needs further clarification; the alteration made is not satisfactory. Please provide more detailed explanation.

2. Line 69, criterion 1 is unclear. Could you please elaborate and specify whether this refers to a culture-positive test or any other diagnostic confirmation method?

3. The content from lines 184 to 188 would be more appropriately placed in the introductory sections of your document. Consider relocating this information for better flow and context.

Author Response

Response to Reviewer 1 Comments

1. Summary

Thank you very much for taking the time to review this manuscript. Please find the detailed responses below and the corresponding revisions highlighted in the re-submitted files.

2. Questions for General Evaluation

Reviewer’s Evaluation

Response and Revisions

Does the introduction provide sufficient background and include all relevant references?

Must be improved

The introduction has been improved as suggested.

Are all the cited references relevant to the research?

Yes

Is the research design appropriate?

Must be improved

The research design and grouping has been revised.

Are the methods adequately described?

Yes

Are the results clearly presented?

Yes

Are the conclusions supported by the results?

Yes

3. Point-by-point response to Comments and Suggestions for Authors

Comments 1: Wang et al. have conducted a commendable preliminary investigation into exosome proteomics analysis to distinguish between NTM and MTB, building upon prior MTB research. This preliminary data holds promise for publication.

However, the absence of validation for the KEGG pathway analysis raises significant concerns about the data's reliability. To solidify your findings, it is essential to perform quantitative western blot or a similar proteomics analysis. Select key proteins from both the downregulated and upregulated categories and confirm your results using an independent secondary method. This may entail substantial work in the lab, but it is crucial for ensuring the quality of your project.

Response 1: First of all, thank you very much for your professional and scientific comments. You have certainly hit the nail on the head with your comment. Our study is the first report on the proteomic analysis of NTM exosomes and is also a pioneering and exploratory screening study. We fully appreciate your concern that independent validation work is necessary to improve the reliability of the study. However, in the case of this study, due to the very low amount of plasma exosomal proteins, our samples were only able us to do screening proteomic studies and not enough for subsequent validation experiments. At the same time, since the prevalence of NTM is very low, it took us a long time to recruit enough cases for the present study. However, we were so eager to make the secrets of NTM exosomes public as soon as possible that we cannot wait for another recruiting period. Next, our research on NTM exosomes will continue, and more cases will be recruited to validate our screened exosome proteomics results. Thank you for pointing this out. We agree with this comment. Therefore, we have added our shortcomings in the discussion section in Line 435-438 on Page 16 in the revised manuscript.

Comments 2: Additional comments:

Throughout your manuscript, you consistently refer to NTM as SGM and RGM collectively, suggesting common findings. However, your study cohorts exclusively involve M. Abscesses and M. Intracellulare. Therefore, it's necessary to narrow your findings to only these two species, as you acknowledged in the introduction. With over 190 NTM species reported, it is not accurate to generalize common findings without supporting data.

Response 2: Your comments are indeed scientific and academic. When designing this study, we did want to compare RGM and SGM, and we grouped them that way in subsequent research stages. However, during the recruitment phase, all the 10 RGM cases we recruited were M. Abscesses and all the 10 SGM cases were M. intracellulare. The probable reasons for may be that M. Abscesses is the most common RGM causing NTM pulmonary disease. And M. intracellulare is the one of the most common SGM causing NTM pulmonary disease in clinical practice. We have narrowed our findings to only these two species in the revised manuscript. The four groups in this study were changed to HC, MAB (M. Abscesses), MAC (M. intracellulare) and MTB groups. And the text, figures and supplemental files throughout the manuscript have been revised accordingly.

Comments 3: Minor comments:

1. Line 45 needs further clarification; the alteration made is not satisfactory. Please provide more detailed explanation.

Response 3: We agree with this comment. Therefore, the content of Line 45 has been further clarified. And more details about the diagnostic potential of exosomes have been explanation.

Comments 4: Line 69, criterion 1 is unclear. Could you please elaborate and specify whether this refers to a culture-positive test or any other diagnostic confirmation method?

Response 4: We agree with this comment. Therefore, the description of Criterion 1 has been expanded to make it clearer. In general, all patients are diagnosed by two positive sputum cultures with the bacterial species well-defined.

Comments 5: The content from lines 184 to 188 would be more appropriately placed in the introductory sections of your document. Consider relocating this information for better flow and context.

Response 5: Thank you very much for your kind advice. After thinking about it, we found that it is more appropriate to put this part of content in the introduction section as you advised. In addition, we've made some clarifications to make the context flow better.

Reviewer 2 Report

The article is deal with the сomparative proteomic analysis of exosomes derived from patients infected with non-tuberculous Mycobacterium and Mycobacterium tuberculosis. The topic discussed is very important for the differential diagnosis of non-tuberculous Mycobacterium from Mycobacterium tuberculosis and for detection of functions of exosomes during infection.

I would like to make a few comments:

1) There are several CD markers to identify exosomes. It is necessary to justify the choice of CD9 and CD81.

2) In the item  Materials, Nanoflow Analysis section, specify the manufacturer of antibodies to CD9 and CD81.

3) In the item  Materials, Extraction of Exosomal Protein please provide a link to the relevant publication for this method.

4) In the item Methods section Statistical analysis (line 155) a mistake was made, and instead of “R-packet” it is necessary to write “R-package”, please provide a link to the relevant publication.

5) In the Discussion item, please, discuss a publication revealing that exosomes derived from M.tb-infected cells induced the pro-inflammatory response of macrophages through TLRs signaling.

Liu M, Wang Z, Ren S, Zhao H. Exosomes derived from mycobacterium tuberculosis-infected MSCs induce a pro-inflammatory response of macrophages. Aging (Albany NY). 2021 Apr 19;13(8):11595-11609. doi: 10.18632/aging.202854

6) In the Discussion item, please, discuss a publication revealing that exosomal ncRNAs may be markers of  Mycobacterium tuberculosis. Discuss and compare the results of your research with the results of mentioned article:

Kaushik AC, Wu Q, Lin L, Li H, Zhao L, Wen Z, Song Y, Wu Q, Wang J, Guo X, Wang H, Yu X, Wei D, Zhang S. Exosomal ncRNAs profiling of mycobacterial infection identified miRNA-185-5p as a novel biomarker for tuberculosis. Brief Bioinform. 2021 Nov 5;22(6):bbab210. doi: 10.1093/bib/bbab210.

Author Response

For research article

Response to Reviewer 2 Comments

1. Summary

Thank you very much for taking the time to review this manuscript. Please find the detailed responses below and the corresponding revisions highlighted in the re-submitted files.

2. Questions for General Evaluation

Reviewer’s Evaluation

Response and Revisions

Does the introduction provide sufficient background and include all relevant references?

Can be improved

The introduction has been improved.

Are all the cited references relevant to the research?

Must be improved

All cited references have been checked and more relevant reference has been added.

Is the research design appropriate?

Yes

The research design and grouping has been revised.

Are the methods adequately described?

Can be improved

The methods have been revised accordingly.

Are the results clearly presented?

Yes

Are the conclusions supported by the results?

Yes

3. Point-by-point response to Comments and Suggestions for Authors

Comments 1: There are several CD markers to identify exosomes. It is necessary to justify the choice of CD9 and CD81.

Response 1: In this study, Nanoflow was used to detect exosome markers, and nanoflow can only detect membrane surface proteins at present. The membrane surface markers commonly detected by exosomes are CD9, CD63, and CD81, which are all transmembrane proteins and are directly involved in the sorting of extracellular vesicle contents. The detection of two of these three markers can help identify exosomes. In this study, CD9 and CD81 were chose to identify exosomes. Thank you for pointing this out. We have added the corresponding content in the revised manuscript.

Comments 2: In the item Materials, Nanoflow Analysis section, specify the manufacturer of antibodies to CD9 and CD81.

Response 2: Both antibodies were brought from BD technology company. Thank you for pointing this out. The manufacturer of antibodies to CD9 and CD81 was added in the revised manuscript.

Comments 3: In the item Materials, Extraction of Exosomal Protein please provide a link to the relevant publication for this method.

Response 3: We agree with this comment. The exosomal proteins used for WB analysis were extracted by the conventional RIPA cleavage method. And a link to the relevant publication for this method has been added in the revised manuscript.

Comments 4: In the item Methods section Statistical analysis (line 155) a mistake was made, and instead of “R-packet” it is necessary to write “R-package”, please provide a link to the relevant publication.

Response 4: Thank you very much for your comment. We agree with this comment. We did make a mistake here. The mistake has been corrected in the revised manuscript. Since not relevant publication could be retrieved, we provided a link to the relevant website.

Comments 5: In the Discussion item, please, discuss a publication revealing that exosomes derived from M.tb-infected cells induced the pro-inflammatory response of macrophages through TLRs signaling.

Liu M, Wang Z, Ren S, Zhao H. Exosomes derived from mycobacterium tuberculosis-infected MSCs induce a pro-inflammatory response of macrophages. Aging (Albany NY). 2021 Apr 19;13(8):11595-11609. doi: 10.18632/aging.202854

Response 5: Thank you very much for your kind advice. You do recommend great publications. The above publication has been discussed in the Discussion item in the revised manuscript as suggested.

Currently, there is still considerable work to be done on the detailed function of exosomes secreted by Mycobacterium-infected cells. A recent study demonstrated that exosomes derived from MTB-infected Mesenchymal stem cells induce a pro-inflammatory response of macrophages through elevating the production of TNF-α, RANTES, and iNOS.

Comments 6: In the Discussion item, please, discuss a publication revealing that exosomal ncRNAs may be markers of Mycobacterium tuberculosis. Discuss and compare the results of your research with the results of mentioned article:

Kaushik AC, Wu Q, Lin L, Li H, Zhao L, Wen Z, Song Y, Wu Q, Wang J, Guo X, Wang H, Yu X, Wei D, Zhang S. Exosomal ncRNAs profiling of mycobacterial infection identified miRNA-185-5p as a novel biomarker for tuberculosis. Brief Bioinform. 2021 Nov 5;22(6):bbab210. doi: 10.1093/bib/bbab210.

Response 6: Thank you very much for your kind advice. You do recommend great publications. The above publication has been discussed in the Discussion item in the revised manuscript as suggested.

Due to limited specimens, only exosomal proteins from Mycobacterium-infected patients were analyzed in the present study. However, both lipids and nucleic acids in exosomes are also very important. Nucleic acids in exosomes, especially non-coding RNAs, are very important in mycobacterial infections. A recent study performed immense parallel sequencing for exosomal ncRNAs and finally collected miR-185-5p as a promising potential biomarker for tuberculosis.

Round 2

Reviewer 1 Report

Authors have strived to address the comments to the best of their ability. However, it is crucial not to impede the findings due to the absence of validation data. I suggest including an editorial note highlighting the limitation regarding the lack of validation in proteomics and suggesting avenues for future research in this area.

Highlight in the introduction that you lacked enough materials to address validation. 

NA

Author Response

Response to Reviewer 1 Comments 1. Summary Thank you very much for taking the time to review this manuscript. Please find the detailed responses below and the corresponding revisions highlighted in the re-submitted files. 2. Questions for General Evaluation Reviewer’s Evaluation Response and Revisions Does the introduction provide sufficient background and include all relevant references? Can be improved The introduction was revised to emphasize that the study was only a preliminary screening study and that our aim was to provide leads for further research. Are all the cited references relevant to the research? Yes Is the research design appropriate? Can be improved We revised our study design to make us our results and conclusions consistent with our study design, while not avoiding the significant shortcomings of our study. Are the methods adequately described? Yes Are the results clearly presented? Yes Are the conclusions supported by the results? Can be improved The results, discussion, and conclusion of the study were revised to make the results, discussion, and conclusions more scientific, as well as to show the shortcomings of the study. 3. Point-by-point response to Comments and Suggestions for Authors Comments 1: Authors have strived to address the comments to the best of their ability. However, it is crucial not to impede the findings due to the absence of validation data. I suggest including an editorial note highlighting the limitation regarding the lack of validation in proteomics and suggesting avenues for future research in this area. Highlight in the introduction that you lacked enough materials to address validation. Response 1: Once again, thank you very much for your professional and scientific comment. As I mentioned in my last response, your comments hit the nail on the head and your advice is scientific and responsible. We couldn't agree more with your suggestion of including an editorial note. It will obviously present the deficiencies of this study to the readers, which is undoubtedly scientific and rigorous. I will follow up with a related suggestion to the editor as well. Regarding your other suggestion of highlighting in the introduction that we lacked enough materials to address validation. I totally agree with you, therefore, I have made some attempted changes as shown below, could you kindly help me to see if they are acceptable? In line 68, the systematic comparative proteomics analysis has been revised as a preliminary screening proteomics analysis, indicating our study is a preliminary study and only screening work was done. In line 70, Our study aimed to reveal the pathogenesis of NTM and MTB infections and the differences between them has been revised as Our study aimed to provide clues for the pathogenesis of NTM and MTB infections. By making this revision, it indicates that our work is only to provide some clues for subsequent research, which is consistent with the essential character of this study. In line 72, we added a highlight that The preliminary results merits validation with large sample to guide clinical practice.
